# Unraveling Chronic Cardiovascular and Kidney Disorder through the Butterfly Effect

**DOI:** 10.3390/diagnostics14050463

**Published:** 2024-02-20

**Authors:** Dimitri Bedo, Thomas Beaudrey, Nans Florens

**Affiliations:** 1Nephrology Department, Hopitaux Universitaires de Strasbourg, F-67091 Strasbourg, France; dimitri.bedo@chru-strasbourg.fr (D.B.); thomas.beaudrey@chru-strasbourg.fr (T.B.); 2Faculté de Médecine, Université de Strasbourg, Team 3072 “Mitochondria, Oxidative Stress and Muscle Protection”, Translational Medicine Federation of Strasbourg (FMTS), F-67000 Strasbourg, France; 3Laboratoire d’ImmunoRhumatologie Moléculaire, INSERM UMR_S 1109, Faculté de Médecine, Fédération Hospitalo-Universitaire OMICARE, ITI TRANSPLANTEX NG, Fédération de Médecine Translationnelle de Strasbourg (FMTS), Université de Strasbourg, F-67000 Strasbourg, France

**Keywords:** cardiorenal syndrome, cardiovascular chronic kidney disorder, butterfly effect, acute injury and long-term aftereffects, vicious circle

## Abstract

Chronic Cardiovascular and Kidney Disorder (CCKD) represents a growing challenge in healthcare, characterized by the complex interplay between heart and kidney diseases. This manuscript delves into the “butterfly effect” in CCKD, a phenomenon in which acute injuries in one organ lead to progressive dysfunction in the other. Through extensive review, we explore the pathophysiology underlying this effect, emphasizing the roles of acute kidney injury (AKI) and heart failure (HF) in exacerbating each other. We highlight emerging therapies, such as renin–angiotensin–aldosterone system (RAAS) inhibitors, SGLT2 inhibitors, and GLP1 agonists, that show promise in mitigating the progression of CCKD. Additionally, we discuss novel therapeutic targets, including Galectin-3 inhibition and IL33/ST2 pathway modulation, and their potential in altering the course of CCKD. Our comprehensive analysis underscores the importance of recognizing and treating the intertwined nature of cardiac and renal dysfunctions, paving the way for more effective management strategies for this multifaceted syndrome.

## 1. Cardiorenal Syndrome (CRS): A Heterogenous Entity

The kidneys, heart, and blood vessels are part of a complex system, where they interact both in healthy states and in disease. When one organ fails, it can affect the others, leading to a condition known as “cardiorenal syndrome”. The definition of this syndrome has evolved over time, highlighting the difficulties in understanding its widespread impact and underlying mechanisms.

Cardiovascular diseases are the main cause of death worldwide, accounting for 31.5% of all fatalities. They result in 4 million deaths every year in Europe alone, with 1.4 million of these deaths occurring in people under 75 years old [1]. Chronic kidney disease (CKD) and heart failure are also prevalent and serious conditions that substantially decrease life quality. It is estimated that CKD affects about 9% of the global population, while heart failure impacts 2% [2,3,4].

These conditions are becoming more common. The prevalence of CKD has risen by 30% from 1990 to 2017, a trend largely linked to an aging population. In 2017, CKD was the cause of 1.2 million deaths globally, and 1.4 million deaths from cardiovascular issues were related to poor kidney function [4]. Forecasts suggest that the need for renal replacement therapy will reach 5.4 million individuals by 2030—twice the number recorded in 2010 [5]. Therefore, enhancing our focus on and research into the interactions between the heart and kidneys—cardiorenal crosstalk—could help avert these grim projections, enabling the early detection and prevention of failure in these vital organs.

### 1.1. Initial Paradigm: A Bidirectional Interaction between Heart and Kidney Dysfunction

Over the past twenty years, various definitions have emerged to clarify the concept of CRS. Initially, CRS was seen as a severe imbalance between cardiac and renal functions, where HF treatment options were limited due to worsening kidney function [6]. In 2008, the Acute Dialysis Quality Initiative (ADQI) broadened this view, introducing a two-way interaction between heart and kidney failures (Figure 1). Ronco and colleagues outlined five subtypes of CRS, categorized by the primary failing organ (either renal or cardiac), whether the failure was acute or chronic, and the role of external factors affecting the heart–kidney interaction [7,8]. Although this classification shed light on potential underlying mechanisms, it has been primarily descriptive and theoretical in nature. Moreover, its application in clinical settings is often complicated by the difficulty in pinpointing which organ’s dysfunction occurred first.

### 1.2. New Paradigm: A Continuum Process That Promotes the Dysfunction of Both Organs

In 2018, the definition of cardiorenal syndrome faced criticism, and a new philosophy regarding this entity was proposed by Zannad and Rossignol [9]. This perspective considers cardiorenal syndrome as a range of related pathophysiological states. Essentially, it is a mutual process that impairs the function of both the heart and kidneys in varying degrees. Research indicates that damage to either the heart or kidneys can trigger a cascade of damage in the other, possibly leading to persistent functional decline in the affected organ. Both heart and kidney diseases often share common risk factors. The progression of fibrosis, or scarring, in either organ can instigate detrimental interactions between the two. Consequently, when both the heart and kidneys malfunction, the outlook can be particularly grim.

### 1.3. Other Classifications

#### 1.3.1. Pliquett Classification

In 2022, Pliquett and colleagues expanded on the ADQI definition of cardiorenal syndrome, building upon the new framework suggested by Zannad and Rossignol [10]. They highlighted the challenge in pinpointing the origin of the initial organ failure, such as in types I or III of cardiorenal syndrome. Pliquett proposed a categorization that differentiates between acute (types I, III, or V) and chronic (types II, IV, or V) conditions. Furthermore, he introduced a distinction between valvular and non-valvular causes, in light of advances in minimally invasive valve procedures like Transcatheter Aortic Valve Implantation (TAVI) or the mitraclip technique. Finally, he recommended including an assessment of the blood volume status to better identify and manage emerging clinical syndromes, such as BRASH syndrome—a condition characterized by bradycardia, renal failure, atrio-ventricular node blockade, shock, and hyperkalemia—often resulting from the combined effects of hyperkalemia and atrio-ventricular node-blocking medications [11].

This method is straightforward for clinical use, but it might be too narrow to fully address CRS. Clinicians often identify CRS merely as severe disturbances in hemodynamics and fluid balance. Treatment typically focuses on using diuretics and ultrafiltration therapy, like the “5B” approach [12], and sometimes includes treatments targeting specific causes—for instance, TAVI for aortic stenosis, pacemakers for bradycardia, or immunosuppressive drugs for dysimmune glomerulonephritis [13,14,15]. However, these strategies often overlook the intricate interplay between the heart and kidneys. By the time such interventions are applied, it may be too late, as options like renin–angiotensin–aldosterone system (RAAS) blockers or gliflozins become less effective once diuretic resistance sets in, necessitating extrarenal replacement procedures. Zannad and Rossignol’s framework aimed to prevent and postpone the onset of these severe stages of CRS. By identifying fibrosis as a key factor, it emphasizes the detection of cardiorenal damage well before organ failure becomes apparent. The pursuit of specific biomarkers could help us overcome the stage of diuretic resistance.

#### 1.3.2. Cardiovascular–Kidney–Metabolic (CKM) Disease: A New Larger Entity

Obesity and metabolic syndrome have become increasingly prevalent worldwide, particularly in industrialized nations [16]. Excess adipose tissue sets off a cascade of detrimental pathways, leading to inflammation, vascular dysfunction, oxidative stress, and insulin resistance. This chronic state is a known contributor to the development of hypertension, type 2 diabetes mellitus, metabolic dysfunction-associated fatty liver disease (MASLD), and atherosclerosis, which are foundational to cardiovascular disease (CVD) and CKD.

In 2023, the American Heart Association (AHA) designated these interconnected health issues as the Cardiovascular–Kidney–Metabolic (CKM) syndrome (Figure 2), a term that encompasses the pathophysiological progression stemming from metabolic syndrome. The AHA outlined a five-stage model to describe the health status within CKM, ranging from Stage 0 (no CKM risk factors) to Stage 4 (clinical CVD in CKM), with kidney disease-associated CVD being identified as the most severe, Stage “4b” [17]. CRS, when associated with metabolic syndrome, falls under the broader CKM syndrome umbrella, with its unique mechanisms.

However, while CRS, CVD, and CKD share similar risk factors and some pathological pathways, metabolic syndrome alone cannot fully explain the intricacies of cardiorenal interactions. CRS represents one of the most serious consequences of CKM, potentially leading to multiorgan dysfunction.

In summary, CKM syndrome outlines a progressive chronic condition that reflects the complex and multi-directional relationships among risk factors, CVD, and CKD. It is crucial to distinguish CRS from CKM as distinct entities. There is a significant need for intensive research to differentiate between the processes that are common to both syndromes and those unique to each [17,18].

**Figure 2 diagnostics-14-00463-f002:**
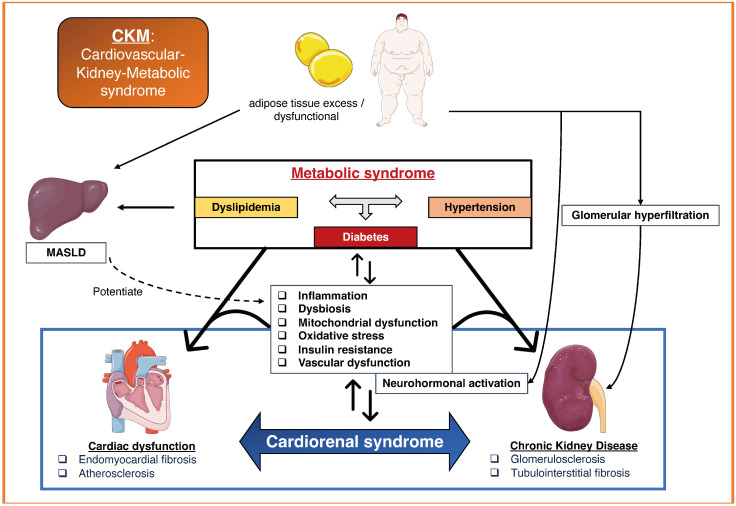
Concept of CKM syndrome, a broad physiopathological entity promoting CRS. Inspired by Ndumele et al. from AHA *Circulation* 2023 [19]. MASLD: metabolic dysfunction-associated steatotic liver disease.

#### 1.3.3. CCKD: Chronic Cardiovascular and Kidney Disorder

In 2024, Zoccali and colleagues, with Zannad and Rossignol among them, argued that the term “Cardiorenal Syndrome” is no longer suitable and advocated for a shift toward the phrase “Chronic Cardiovascular and Kidney Disorder” (CCKD) [19]. They posited that “syndrome” is a restrictive term, often associated primarily with diuretic resistance and the management of organ failure. They emphasized the need to highlight the origins of cardiovascular and kidney damage and the biological changes that occur early on. The term “disorder” is preferred as it underscores the integration of kidney pathology within the cardiovascular domain, acknowledging the shared risk factors and pathophysiological processes between CVD and CKD.

The authors have updated the term “Cardiorenal syndrome revisited” to “CCKD” to encapsulate the concept of a continuum. In this continuum, kidney dysfunction can hasten the progression of cardiovascular disease and vice versa. The initial presentation of CCKD does not always involve simultaneous cardiac and renal failure. Clinically, CCKD might predominantly present as CVD or display the typical signs of CKD. Moreover, even if one organ appears to be functioning well, it may still have subclinical damage.

Major adverse cardiovascular events (MACEs) are often used as the endpoint in randomized clinical trials focusing on cardiovascular outcomes. However, a well-defined primary composite endpoint that includes kidney events is lacking [20,21,22]. Kidney function is usually considered only in terms of safety outcomes. By embracing a CCKD framework, there is an opportunity to consider both cardiovascular and kidney events as composite endpoints in future research [23].

## 2. From Epidemiology to a Vicious Cycle of Cardiorenal Interaction

### 2.1. Heart and Kidney Disease: A Perilous Connection

CVD is not only the most frequent complication of CKD, but also the leading cause of death, responsible for over 50% of fatalities in patients with CKD [24]. This association is even more stark among those with end-stage renal disease (ESRD). For instance, in the HEMO study—a randomized multicenter trial involving 1846 patients undergoing chronic hemodialysis—80% had some form of heart disease, with 39% suffering from ischemic heart disease (IHD), 40% with congestive HF, and 31% experiencing arrhythmias. 75% of those with end-stage renal disease exhibit left ventricular hypertrophy (LVH), and 40% concurrently had coronary artery disease [25]. Moreover, the incidence of chronic heart failure is 15–20 times greater in individuals with renal insufficiency compared to the healthy population [26,27,28,29]. Microalbuminuria and macroalbuminuria are recognized not only as CVD risk factors, but also as independent predictors of HF [30].

These findings have led to the recognition of uremic cardiomyopathy, which shares common risk factors with CVD and has additional specific factors like uremic toxins, anemia, hypervolemia, oxidative stress, inflammation, insulin resistance, and Chronic Kidney Disease–Mineral and Bone Disorder (CKD-MBD) [31,32,33]. Kidney transplantation has been found to lower the risk of significant cardiovascular events and improve the left ventricular ejection fraction, thus supporting the notion of kidney disease-specific cardiomyopathy promoters [34].

Up to 30% of patients with CKD suffer from chronic HF, and conversely, around 40–50% of HF patients have coexisting renal dysfunction [35]. Notably, about 20% of heart failure patients have moderate to severe renal insufficiency, with over 60% showing at least mild renal insufficiency [36].

It has been observed that as HF worsens, there is a corresponding decline in kidney function. The CRIC study indicated that a history of HF among patients with CKD was linked to a 29% higher risk of progressing to ESRD or experiencing a 50% reduction in the estimated glomerular filtration rate (eGFR), compared to those without HF [37]. The Digoxin Intervention Group trial, which assessed the efficacy of digoxin in patients with stable heart failure and a left ventricular ejection fraction above 45% in sinus rhythm, found that 46% of participants had an eGFR lower than 60 mL/min/m^2^, with mortality inversely related to eGFR. This relationship was not linear, showing a sharp increase in risk below an eGFR threshold of approximately 50 mL/min/m^2^ [38].

While chronic interactions between heart and kidney dysfunction are somewhat easier to define, the epidemiology of acute cardiorenal injury is more complex, mainly due to the difficulty in pinpointing the initial organ failure, especially in CRS types 3 and 5. The heterogeneous nature of acute kidney injury (AKI) further obscures the precise prevalence rates [39]. Nevertheless, Odutayo et al. presented in a systematic review and meta-analysis that AKI was associated with an 86% higher risk of cardiovascular mortality and a 38% higher risk of major cardiovascular events. Specifically, AKI correlated with a 58% increased risk of heart failure and a 40% increased risk of acute myocardial infarction. These heightened risks persisted even when considering the severity of AKI and the proportion of adults with underlying ischemic heart disease [40]. Although the data are somewhat limited and retrospective, potentially introducing biases, they corroborate the renocardiac interplay and the concept of a continuum. The current literature still lacks information on the emergence of cardiac injury markers during episodes of AKI.

### 2.2. Focus on Cardiorenal Syndrome Type 1

#### 2.2.1. Definition and Peculiarity

CRS type 1 is defined by the onset of AKI in patients experiencing acute cardiac conditions, particularly acute decompensated heart failure (ADHF). It has been observed that between 25 and 50% of patients with ADHF develop AKI, often subsequent to ischemic or non-ischemic heart disease [41], Additionally, 30–50% of AKI cases occur following cardiac surgery or valvular reconstruction [42,43,44,45]. A retrospective analysis of 118,465 patients hospitalized with decompensated heart failure revealed that only 9% had normal kidney function upon admission, while 56% had an eGFR between 60 and 15 mL/min/m^2^ [46].

ADHF is a unique condition, distinct from acute cardiac injury without heart failure, primarily because it usually develops from an existing cardiac disorder. This highlights the heterogeneity of CRS type 1, suggesting that it is more reflective of a consequence of an ongoing cardiorenal disorder in ADHF cases.

Cardiogenic shock (CS), fitting the profile of CRS type 1, has a well-established treatment protocol. Its renal complications are mainly due to hemodynamic imbalances, stemming from low cardiac output and elevated central venous pressure. The incidence is quite high, exceeding 30–50% depending on various definitions [47]. In a Danish study involving 5032 cardiogenic shock patients, 13% experienced severe AKI, requiring Renal Replacement Therapy (RRT). This condition significantly raised mortality risk (HR 15.9), with a 64% hospital mortality and 43% five-year mortality among survivors (HR 1.55) [48]. The severity of cardiogenic shock, akin to other shock states, underscores the broad spectrum of this CRS subtype. Its long-term consequences, such as an increased risk of hemodialysis, point toward the progressive development of pathological cardiorenal pathways. Moreover, recent machine learning studies have identified three phenotypes of CS: “non-congestive”, “cardiorenal”, and “cardiometabolic”. The cardiorenal phenotype is strongly associated with adverse prognostic events [49,50].

#### 2.2.2. Myocardial Infarction Associated with AKI

##### Epidemiology and Prognosis

Myocardial infarction (MI) associated with AKI occurs in approximately 10–30% of cases and is linked to a long-term poor prognosis, even in cases of mild AKI. This suggests the possibility of a harmful cycle following a single cardiorenal event [51,52]. AKI is known for its high mortality rate, which ranges from 30 to 50% in severely affected patients. Notably, this increased mortality risk continues for up to five years after the event, especially for those who suffer persistent AKI during hospitalization [53,54]. An extensive observational study utilizing the Veterans Affairs database, encompassing 36,980 patients, illustrated that United States veterans with AKI alone faced worse outcomes compared to those who had suffered a myocardial infarction without AKI. The highest mortality rate was observed in the group with both MI and AKI (57.5%), whereas the lowest mortality rate (32.3%) was found in patients who had an uncomplicated MI admission [55]. These data underscore the severe impact of AKI on patient outcomes, particularly when it occurs in conjunction with myocardial infarction.

##### MI Early Phase and Controversial Contrast-Induced Nephropathy

Some research has shown that the prognosis of AKI associated with MI varies significantly depending on when AKI occurs. The first three days post MI are considered the early phase, while days four to seven are the late phase. The all-cause and cardiovascular mortalities are specifically linked to AKI that begins in the early phase of MI [56]. Initially, it was hypothesized that late-phase onset might be due to excessive diuretic use, while early-phase onset was thought to be related to contrast-induced nephropathy (CIN) [57,58]. However, this view is contentious, especially if it leads to delaying crucial cardiac interventions.

The concept of CIN is a subject of ongoing debate. While contrast media are theoretically nephrotoxic, the impact of cardiorenal syndrome often outweighs the risks associated with restricting diagnostic or therapeutic procedures. For example, despite the risk of AKI, patients often experience an improvement in eGFR following a TAVI procedure. The dosage of contrast media has been linked to AKI, but the duration of the procedure can be a confounding factor, possibly indicating endothelial dysfunction related to the catheterization procedure [59].

Kosaki et al. reported that patients who developed AKI in the early phase of MI had higher levels of C-reactive protein (CRP), suggesting that an inflammatory response in the cardiorenal axis might be a more significant contributor to AKI and long-term adverse events [60]. On the other hand, recent data, particularly from studies using a “zero-contrast” strategy, seem to support the notion that contrast media are not as toxic to the kidneys as previously thought, as evidenced by similar rates of AKI in these scenarios [61,62]. This growing body of evidence suggests that the relationship between contrast media and AKI may be less direct than previously believed.

##### Prognostic Impact Amidst Diverse AKI Definitions

The definition of AKI has varied across studies, with different criteria set by groups like AKIN, RIFLE, and, more recently, KDIGO. In cardiological research, kidney injury is often less stringently defined, using the term Worsening Renal Function (WRF). To address specific scenarios like increases in creatinine levels following the initiation of renin–angiotensin–aldosterone system (RAAS) blockers, SGLT-2 inhibitors (gliflozins), or post-decongestion therapy, the term “Pseudo-WRF” was introduced [63,64].

WRF is typically defined as an absolute serum creatinine increase of more than 0.3 mg/dL or a 25% rise from the baseline, although KDIGO’s definition is slightly different (0.3 mg/dL increase within 48 h, a 1.5× increase in creatinine from the previous 7 days, or urine output less than 0.5 mL/kg/h for 6 h). Despite these variations in defining AKI, “True-WRF” still shows a strong association with increased mortality [65].

In summary, the epidemiology of CRS type 1 demonstrates variability on multiple levels. Nonetheless, there is a consistent and clinically significant correlation with a poor prognosis. Notably, even a minor increase in creatinine levels, as small as 0.1 mg/dL, is associated with worsened outcomes, even after adjusting for known confounding factors [66]. Such adverse cardiorenal events propel patients into a detrimental cycle, synergistically accelerating renal dysfunction and left ventricular remodeling [67,68].

### 2.3. CCKD Definition: The Vicious Cycle of Cardiorenal Interaction

Zannad and Rossignol have highlighted that while classifying conditions as cardiorenal or renocardiac syndromes may be convenient for epidemiological and clinical categorization, this approach might be too simplistic and late-stage to fully capture the underlying mechanisms of the condition.

Recent large-scale randomized controlled trials have brought attention to current therapeutic strategies. These include sodium-glucose cotransporter 2 (SGLT2) inhibitors [69,70,71,72,73,74], RAAS blockers [75,76,77,78], angiotensin receptor neprilysin inhibitors (ARNI) [79,80], mineralocorticoid receptor (MR) antagonists [81,82,83], and glucagon-like peptide 1 (GLP1) ± glucose-dependent insulinotropic peptide (GIP) receptor agonists [84,85]. These treatments show promise in preventing major adverse cardiac and kidney events (MACEs and MAKEs). Although the exact mechanisms of these therapies are still under investigation, they are known to target various pathways involved in the development of cardiorenal fibrosis (Figure 3).

The coexistence of chronic heart disease and chronic kidney disease is frequent, and each condition can worsen the other.

The 2008 ADQI definition fails to address the chronic effects on one organ following an acute injury to the other. For example, it does not detail how an acute kidney injury episode can lead directly to chronic heart failure development or how an acute cardiac event can lead to CKD without necessarily progressing to chronic heart failure (Figure 3).

Halimi and colleagues’ nationwide study involving 385,687 French patients with cardiorenal syndrome aimed to examine long-term major outcomes, irrespective of the chronology of CRS onset [86]. In their study on 5,123,193 patients, 84.0% had HF, 8.9% had CKD, and both groups showed similar demographics and prevalences of diabetes and hypertension. Of these, 7.1% had CRS, with subtypes including cardiorenal (44.6%), renocardiac (14.5%), and simultaneous CRS (40.8%). A supplementary study by the same researchers found that among patients with CRS, half were diagnosed with diabetes mellitus. Additionally, the presence of type 2 diabetes mellitus was associated with a doubling of the risk for CRS and tended to affect patients at a younger age [87].

All CRS phenotypes had a higher risk of all-cause death, cardiovascular death, and heart failure compared to patients with isolated CKD or HF, with the worst outcomes in the cardiorenal group. However, minimal outcome differences were observed among CRS phenotypes, indicating that the sequence of heart failure and CKD onset does not significantly affect cardiovascular, cerebrovascular, and kidney outcomes. The study also found that the coexistence of HF and CKD factors additively, rather than multiplicatively, increases the risk of mortality and adverse cardiovascular outcomes.

This finding challenges the traditional view of CRS subtypes as mere risk markers and suggests that the chronology of CRS may have distinct long-term impacts on major endpoints.

## 3. Cardiorenal Butterfly Effect

### 3.1. Definition of the Concept

Based on these insights, we propose a novel concept to deepen our understanding of CCKD. This concept, which we call the “butterfly effect”, encompasses not only the chronic aspects of the disorder, but also incorporates the impact of acute organ insults that may lead to the development of long-term significant organ damage. The butterfly effect, in this context, suggests that an acute, often resolvable alteration in one organ (either cardiac or renal) can trigger significant long-term effects. This is particularly relevant when considering that even after the recovery of the affected organ, the initial insult might set off pathological biological pathways, potentially leading to dysfunction in the other organ (Figure 4).

The manifestation of dysfunction can vary based on the individual’s risk factors, potentially occurring in the following:■The initially affected organ, particularly if it does not fully recover or if there are cardio/nephrotoxic conditions involved.■The other remote organ, especially if the initially affected organ recovers but there are predominant toxic factors impacting the other organ.■Both organs simultaneously, if the initial insult is severe enough to trigger a rapid and harmful cycle of mutual dysfunction.

This interpretation of the natural course of cardiorenal dysfunction following an acute organ insult is theoretical and highly individualized, considering the complexity of interacting pathological states. Moreover, an individual might experience multiple episodes of acute cardiac or renal insults.

The cardiorenal butterfly effect is not a necessary condition for the development of clinically significant cardiovascular and kidney outcomes because of the following:■It is not required, as CKD or HF can evolve into CRS without an acute insult. However, it is important to acknowledge that even a reversible, acute renal or cardiac insult could have underappreciated long-term consequences.■It is not sufficient on its own, depending instead on the presence of metabolic and systemic risk factors. These factors play a crucial role in initiating cardiorenal crosstalk and, subsequently, cardiorenal fibrogenesis.

From a clinical and research perspective, identifying key drivers of this pathological cycle is vital. This includes recognizing specific biomarkers and implementing targeted therapies for high-risk individuals. Such an approach aims at early detection and intervention, potentially mitigating the progression of CCKD.

The onset of CKD after an acute cardiac insult characterizes what we term the type 1 cardiorenal butterfly effect. Conversely, the evolution of cardiac remodeling leading to HF following an episode of AKI delineates the type 2 cardiorenal butterfly effect.

### 3.2. Cardiorenal Butterfly Effect Type 1

#### 3.2.1. Description

The prevalence of CRS is estimated to be around 7% of the population. CKD following an acute cardiac injury defines “type 1 butterfly effect”. In this scenario, an acute cardiac event—such as a reperfused myocardial infarction, myocarditis, or treated arrhythmogenic heart disease—that is potentially curable could lead to the development of subclinical renal lesions. These lesions might not be immediately apparent but could eventually progress to CKD.

On the other hand, a more severe and challenging acute cardiac event, like a massive anterior myocardial infarction with delayed intervention, could result in heart failure. This situation may occur with or without concurrent renal failure. Such instances are classified as simultaneous CRS. In these cases, the acute cardiac insult is so significant that it precipitates a direct and immediate impact on both the heart and kidneys, reflecting a more synchronized dysfunction in the cardiorenal continuum.

#### 3.2.2. Clinical Evidence Supporting the Concept

The link between acute cardiac events and the subsequent risk of CKD and ESRD has been relatively understudied. However, existing research indicates that conditions like heart failure or coronary artery disease can accelerate the progression to ESRD, as observed in retrospective cohort studies [88,89,90].

A significant retrospective study in Sweden, the Stockholm CREAtinine Measurements (SCREAM) project, offers insights into what we describe as the “type 1 butterfly effect” induced by acute cardiac events [91]. This study analyzed the rate of estimated glomerular filtration (eGFR) decline before and after hospitalization due to HF, coronary heart disease (CHD), or stroke. The study excluded creatinine levels 3 months before and after the event to avoid confounding factors such as the impact of therapies like decongestion or the introduction of RAAS blockers or SGLT2 inhibitors, which could influence GFR but also potentially improve long-term prognosis [92,93]. The findings showed a significant acceleration in eGFR decline post acute cardiac event, especially in HF and CHD. Moreover, patients with an eGFR ≥ 60 mL/min/1.73 m^2^ had the higher eGFR decline, almost 200% faster post event, indicating an important role for these conditions in the pathophysiology of cardiorenal syndrome. Further, the same team assessed the risk of ESRD following cardiac events in the Atherosclerosis Risk in Communities (ARIC) database that collected the data prospectively. This study included over 9000 individuals without prevalent cardiovascular disease. The analysis revealed that incident heart failure was the most significant risk factor for Kidney Failure Replacement Therapy (KF-RRT), followed by coronary heart disease. The cumulative incidence of ESRD among participants with heart failure was approximately 10% at 5 years and 15% at 10 years [92].

Recently, Mark et al. conducted a comprehensive study involving over 25 million participants from the CKD Prognosis Consortium. This study assessed the impact of prevalent and incident CHD, stroke, HF, and atrial fibrillation (AF) on the outcomes related to KFRT. They found that both prevalent and incident CVD were associated with increased risk of KFRT, with the highest hazard ratios observed in the first three months post CVD incidence, particularly following HF hospitalizations, demonstrating an extremely strong temporal association [93].

These studies collectively underline the importance of acute cardiac events as potential initiators of a detrimental cycle leading to chronic cardiovascular and kidney disorders. Understanding and exploring the mechanisms behind this “type 1 butterfly effect” is crucial for prevention, early recognition, and treatment strategies in Chronic Cardiovascular and Kidney Disorder.

#### 3.2.3. Biomarkers

The identification of the type 1 butterfly effect in CRS often encounters delays and presents challenges. Nevertheless, emerging biomarkers for renal damage could play a crucial role in identifying populations at high risk and preventing the progression of this cardiorenal path. Biomarkers such as kidney injury molecule-1 (KIM1), Neutrophil Gelatinase-Associated Lipocalin (NGAL), N-Acetyl-Beta-D-Glucosaminidase (NAG), and cystatin-C for estimating GFR are under study, with mixed diagnostic outcomes noted [65,94]. KIM-1, a glycoprotein detected in higher urinary concentrations post AKI and during CKD development, correlates with fibrosis in animal studies [95,96,97]. Its role extends from protective anti-inflammatory effects in AKI to associations with fatty acid uptake in CKD, particularly in diabetic kidney disease [98,99,100]. NGAL, a glycoprotein, may act as an inflammation modulator and is considered a potential biomarker for AKI and CKD [101,102,103,104], despite reduced specificity in conditions like anemia or sepsis. Urinary NAG, indicative of proximal tubular cell damage, shows promise in acute and chronic renal pathologies [105,106]. Proenkephalin A (PENK) [107,108] and liver fatty acid-binding proteins (L-FABP) [109], linked to acute heart failure and ischemic tubular injury, respectively, could predict kidney dysfunction and mortality. These biomarkers warrant further robust, longitudinal studies post acute cardiac events to enhance the clinical detection of cardiorenal effects.

#### 3.2.4. Preclinical Evidence Supporting the Concept

The long-term impact of a cardiac insult on kidney function has been a data gap over the past decades. However, in 2004, a model of myocardial infarction in unilaterally nephrectomized rats revealed that glomerulosclerosis and proteinuria were exacerbated by myocardial infarction, even with a small infarct size [110]. Lekawanvijit et al.’s study provided a pioneering exploration of induced nephropathy in a rat model following MI, compared to a sham procedure [97]. In this research, they observed a decrease in glomerular filtration rate after MI at 1 and 4 weeks, with no significant changes at 8 and 12 weeks, but a further decline at 16 weeks. Notably, increased IL-6 gene and transforming growth factor (TGF)-β protein expression, as well as macrophage infiltration in the kidney cortex, were evident at 1 week. There was a significant increase in renal cortical interstitial fibrosis from 4 weeks onward, with TGF-β bioactivity consistently elevated. The peak of fibrosis was noted at 16 weeks, suggesting an initial hemodynamic AKI in this severe MI model, followed by nephropathy likely driven by pro-fibrotic and inflammatory interactions, culminating in renal fibrosis. This progression could be indicative of the “butterfly effect type 1”. Moreover, kidney injury molecule-1 positivity in tubules was more pronounced in MI animals, peaking at 1 week, suggesting its potential as a biomarker for further investigation.

Additional studies have emphasized the role of inflammation as a mediator in heart–kidney crosstalk. After MI, animal research indicates a systemic inflammation characterized by elevated levels of activated monocytes CCR2+ in peripheral blood. Moreover, there is evidence of leukocyte recruitment, particularly macrophages, at remote sites in both the myocardium and renal parenchyma. This recruitment is accompanied by the overexpression of leukocyte adhesion molecules such as VCAM-1 and pro-inflammatory cytokines like TNF-α and IL-6 genes in kidney tissue. MI-induced consequences also involve endothelial dysfunction, tubular cell apoptosis associated with increased NGAL biomarker expression, and renal interstitial fibrosis. Notably, these effects were alleviated with liposome clodronate, a systemic macrophage-depleting agent [111,112]. One translational study using whole-body positron emission tomography (PET) with the CXCR4 ligand 68Ga-Pentixafor in mice post MI, demonstrated a pathological cardiorenal connection [113]. Early transient myocardial CXCR4 upregulation was observed after MI, with direct correlation between cardiac and renal PET signals over time. In humans, renal CXCR4 signals correlated with signals from infarcted and remote myocardial areas and were independently associated with adverse renal outcomes.

Furthermore, the role of LOX-1, a pro-inflammatory and pro-oxidative molecule, was studied in mice lacking this receptor (LOX1 KO) subjected to MI. These mice showed fewer structural alterations and improved cardiac and renal function compared to wild-type mice suggesting the potential role of LOX-1 for leukocyte accumulation in ischemic and injured cardiorenal tissues [114].

Another study focused on apela, a hormone acting on the APJ receptor, in a murine MI model [115]. Apela was found to reduce renal fibrosis, inflammation, apoptosis, and DNA damage response in vitro and in vivo, thus reducing renal tubular lesions and dysfunction induced by ischemia-reperfusion injury [116]. Mice treated with exogenous apela post MI showed improved renal and cardiac parameters including reduced tissue inflammatory markers, suggesting its potential as a therapeutic agent against cardiorenal syndrome following MI.

Although these animal models provide important information, their direct relevance to clinical settings, especially in cases of non-revascularized myocardial infarctions (MIs), is somewhat constrained. Presently, research initiatives, including our own lab’s work, are concentrated on coronary ischemia-reperfusion models. This type of model is aimed at a more in-depth exploration of nephropathy and its underlying mechanisms in scenarios where swift reperfusion is the prevailing treatment standard.

### 3.3. Cardiorenal Butterfly Effect Type 2

#### 3.3.1. Description

In the “type 2 butterfly effect” scenario, HF develops following AKI. In this situation, a potentially treatable acute renal event, such as renal infarction, acute tubular necrosis, or malignant nephroangiosclerosis, can initiate subclinical cardiac lesions that may eventually lead to HF. Furthermore, 40% of CRS cases exhibit a simultaneous sequence of events. This simultaneous occurrence could be the result of a combined type 1 + 2 butterfly effect, often seen in conditions like sepsis.

#### 3.3.2. Clinical Evidence Supporting the Concept

There is significant evidence supporting the idea that AKI is associated with delayed and far-reaching consequences, such as increased blood pressure, cardiovascular events, the development and progression of CKD, ESRD, and higher mortality rates [117,118]. This connection is evident even after a five-year follow-up period, as shown in a population-based cohort study. This study revealed that both early- and late-onset AKI within 30 days of elective cardiac surgery were linked to an increased five-year risk of myocardial infarction, heart failure, stroke, and higher all-cause mortality [119].

For instance, Lu et al. analyzed data from 3296 COVID-19 patients hospitalized with AKI. They categorized the AKI recovery into early (<48 h), delayed (2–7 days), and prolonged (>7–90 days) categories. The study found that 28.6% of COVID-19 patients experienced AKI, with 58.0% showing early recovery, 14.8% delayed recovery, and 27.1% prolonged recovery. Longer AKI recovery times correlated with a higher prevalence of CKD. Notably, many COVID-19 patients developed MAKEs, recurrent AKI, and new-onset CKD within 90 days, especially those in the prolonged recovery group. The incidence of MACEs peaked 20–40 days post discharge, while MAKEs peaked 80–90 days post discharge. Although there are some biases, this study exemplifies what we refer to as the cardiorenal butterfly effect type 2, where hospital AKI in COVID-19 survivors (a minor cause) leads to significant cardiovascular and kidney outcomes (substantial consequences) [120]. Further investigations have focused on biomarkers during and post AKI to understand the underlying pathological interplay. Soluble receptors of tumor necrosis factor, known to be associated with cardiovascular adverse events and CKD progression, are independently elevated and persist even three months post AKI among patients who developed HF. These biomarkers could serve as potential clinical targets [95,121,122,123,124,125].

Additionally, Odutayo et al.’s work supports this type 2 butterfly effect, showing a 58% increased risk of heart failure and a 40% increased risk of acute myocardial infarction following AKI [40]. This underlines the complexity of the cycle and the potential for inducing an acute insult on the other organ, leading to a further pathogenic acceleration through a type 1 butterfly effect.

Recent retrospective studies reinforce these findings. A propensity score-matched cohort study involving 471,176 patients hospitalized with AKI (and matched with an equal number without AKI) found that AKI was associated with higher rates of rehospitalization for various reasons, including end-stage renal disease, heart failure, myocardial infarction, and volume depletion at 90 and 365 days post discharge. The mortality rate was also higher in the AKI group, highlighting the risk elevation and the butterfly effect induced by AKI [126].

A recent study employing cardiac magnetic resonance imaging (MRI) revealed that individuals who survived critical illnesses, with no prior history of cardiac conditions, showed evidence of myocardial fibrosis and systolic dysfunction. Notably, these cardiac abnormalities were more marked in patients who had experienced severe AKI [127]. This correlation is currently being explored in a forthcoming study, “Multiparametric Cardiac MRI for the Detection and Quantification of Myocardial Injury Following Acute Kidney Injury” (NCT 05034588). This investigation is at the forefront of using medical imaging to elucidate the “type 2 butterfly effect” in cardiology.

#### 3.3.3. Preclinical Evidence Supporting of Concept

Martin et al. discovered that uninephrectomy in rats, even without evident chronic kidney disease or heart failure, led to left ventricular myocardial fibrosis and increased apoptosis in cardiomyocytes. Notably, there were no changes in GFR, proteinuria, or plasma brain natriuretic peptide levels compared to sham-operated rats. Gene microarray analysis of the left ventricular wall showed dysregulation in a broad spectrum of genes related to apoptosis and fibrosis, particularly in the TGF-ß pathway, at both 4 and 16 weeks post uninephrectomy [128]. Additionally, renal failure in rats correlated with a heightened vulnerability of cardiomyocytes to ischemia-reperfusion injury. This increased risk of cardiac complications post ischemia-reperfusion injury was also linked to changes in the adiponectin levels and signaling pathways [129]. These findings provide early evidence supporting the concept of a type 2 butterfly effect, where renal impairment can predispose the heart to increased injury.

In animal models, particularly those involving ischemia reperfusion, AKI has been associated with an increase in various inflammatory cytokines, including interleukin 1 and 6, intercellular adhesion molecule 1, interferon gamma, and tumor necrosis factor alpha [130]. These cytokines, especially TNF-alpha, have recently been identified as key players in cardiac remodeling in CKD, based on single-nucleus analysis of left ventricular wall cells in experimental CKD models in mice [131]. Earlier studies have shown that blocking TNF-alpha can improve cardiac outcomes following AKI [132]. Additionally, AKI has been linked with cardiac mitochondrial abnormalities and changes in the heart’s metabolomic profile [133,134], indicating a complex interaction between kidney injury and cardiac health.

Prud’homme et al. recently emphasized the critical role of Galectin-3 in remote cardiac remodeling following AKI [135]. They demonstrated that cardiac remodeling is specifically induced by the kidney’s inflammatory response, as shown by the presence of cardiac remodeling in a unilateral ureteral obstruction model, in contrast to limb ischemia reperfusion. This study also found a positive correlation between Galectin-3 levels in ICU patients and the severity of AKI. This aligns with other research indicating a link between Galectin-3 levels and adverse cardiac outcomes in both cardiac and kidney diseases [136,137,138].

Galectin-3, a member of the beta-galactosidase-binding lectin family, is secreted by activated macrophages and plays a role in fibrosis by promoting collagen deposition in the extracellular matrix at both renal and cardiac levels. Its significance has recently expanded as a prognostic marker for cardiorenal syndrome, particularly in heart failure populations with reduced ejection fraction [139,140]. Interestingly, Galectin-3 regulation may be influenced by cardiotrophin-1 (CT-1), a cytokine with diverse tissue and cellular effects [141]. While CT-1 generally promotes cell survival, its prolonged overexpression can lead to worsened cardiac hypertrophy and fibrosis [142]. The effect of CT-1 on renal tissue is debated; initially thought to mediate renal fibrosis [143], recent studies suggest it may protect against renal damage in certain models of kidney obstruction [144,145].

The IL33/ST2 pathway plays a crucial role in the type 2 butterfly effect’s pathophysiology [146,147]. IL33, an alarmin released by epithelial cells during various injuries, exerts pro-inflammatory effects and is implicated in increasing AKI severity and kidney inflammation [148,149,150]. Contrarily, IL33 has shown protective effects in heart failure and myocardial infarction [151,152]. However, some studies associate IL33 with worse cardiac outcomes, particularly post myocardial infarction [153]. Our research demonstrates that IL33 directly contributes to cardiac remodeling post AKI through the ST2 receptor on cardiomyocytes [154]. IL33 overexpression in mice led to a detrimental cardiac phenotype, whereas blocking IL33 during AKI protected the heart. Although IL33 secretion is short-lived, its impact on ST2 receptor expression is sustained, influencing long-term cardiac remodeling. The soluble ST2 (sST2), part of this axis, correlates with adverse cardiovascular outcomes in cardiac and renal conditions [155,156,157,158,159]. While traditionally viewed as a decoy receptor, emerging evidence suggests sST2’s intrinsic pathological role [160].

## 4. Therapeutic Approach for Alleviating Butterfly Effect

### 4.1. Potential Therapies in Cardiorenal Medicine

Cardiorenal syndrome poses a significant challenge in modern societies, particularly in Western populations, where this interrelated condition is becoming increasingly prevalent. Advances in diagnosing and treating conditions like myocardial infarction have improved patient prognoses, reducing the risk of severe heart failure and subsequent kidney dysfunction. RAAS inhibitors were among the first therapies to highlight the integrated physiology of the heart and kidney continuum [76,78,161,162,163,164]. These treatments are now fundamental in managing cardiorenal conditions like post-myocardial infarction, heart failure, and chronic kidney disease. The future may see RAAS inhibition through RNA Interference Therapeutic Agents, currently explored for hypertension and possibly beneficial for cardiorenal crosstalk [165]. However, their role in protecting against type 2 butterfly effect in acute kidney injury is still under investigation [166,167,168].

Recently, new therapies have emerged for treating cardiorenal and broader cardio–kidney–metabolic syndromes, largely driven by metabolic factors. SGLT2 inhibitors along with RAAS blockade have become essential in treating cardiac and kidney diseases [69,71,72,73,74,169,170,171,172]. The effectiveness of GLP1 agonists in managing metabolic control and thereby improving cardiorenal pathologies is also notable [70,84,85,173,174,175,176], particularly as insulin resistance and metabolic disturbances are key in the fibrogenesis pathways that drive cardiorenal syndrome progression. Finerenone and other mineralocorticoid antagonists, beneficial in reducing post-myocardial infarction fibrosis and slowing chronic kidney disease progression, are also promising [81,82,83,177,178,179]. Their precise roles in cardiorenal protection are yet to be fully determined. Aldosterone and aldosterone synthase inhibitors also seem to be promising for the treatment of resistant hypertension and may offer additional benefits in CCKD [180,181].

Potential future treatments may target specific pathways like Galectin-3 inhibition or IL33/ST2 pathway modulation, but strong evidence for their effectiveness is still emerging.

### 4.2. Addressing the Butterfly Effect in Current Cardiorenal Trials

Recent and ongoing cardiorenal trials, such as the EMMY and DAPA-MI (NCT03087773), which evaluate the effectiveness and safety of empagliflozin and dapagliflozin following MI, unfortunately overlook key renal parameters like albuminuria or changes in creatinine levels over time—essential markers of renal injury [182,183]. This oversight is also seen in other significant randomized controlled trials, such as the CANTOS and COLCOT, which explored the potential of innovative, possibly anti-fibrotic medications but failed to consider renal outcomes, even as adverse events [184,185].

In clinical settings, there is a noticeable reluctance among physicians to prescribe cardioprotective or nephroprotective treatments immediately after an acute incident, due to fears of precipitating renal adverse events. However, evidence increasingly supports the contrary, advocating for the early adoption of treatments that safeguard both heart and kidney functions post acute injuries. For example, the PIONEER-HF trial delved into the efficacy and safety of sacubitril-valsartan for patients with heart failure with reduced ejection fraction, who had been admitted for acute decompensated heart failure. Although the primary endpoint, based on N-terminal pro–B-type natriuretic peptide levels, may be contentious, the trial’s results offer promising insights, hinting at potential benefits for both cardiac and renal health. Moreover, it reported no increase in the incidence of WRF, suggesting a manageable risk for this severely impacted patient group [186]. Pertinently, within the scope of the type 2 butterfly effect, studies have shown that prescribing RAAS inhibitors to patients discharged from the ICU after surviving AKI is linked with reduced one-year mortality rates [187]. These observations are highly encouraging, merit further investigation, and highlight the critical need for interventions designed to prevent cardiovascular complications following AKI, such as post-AKI RAAS blocker therapy, as recommended by Legrand and Rossignol [188].

## 5. Limitations and Challenges with the Cardiorenal Butterfly Effect Approach

The cardiorenal butterfly effect concept is promising but also fraught with complexities. Within pre-clinical research, the primary challenge lies in refining animal models to mirror the clinical landscape, thereby facilitating the translation of findings from the bench to bedside. The prevalent use of renal ischemia reperfusion as an animal model for AKI may not adequately represent clinical conditions, where acute tubular necrosis is a predominant cause. This disparity raises concerns about the direct application of preclinical findings to patient care. Conversely, models of coronary ischemia reperfusion align more closely with clinical scenarios, though the nuances of revascularization procedures are not easily replicated in rodent models.

Translating the butterfly effect into routine clinical practice requires a paradigm shift in the current standard of care. Many AKI patients, even those with severe conditions, may not consult nephrologists, presenting an even greater challenge for cardiologists to assess cardiovascular risks in these patients. Similarly, nephrologists may not routinely evaluate the implications of acute cardiac conditions on chronic kidney disease risk.

The timely diagnosis of AKI is pivotal for the clinical adoption of the butterfly effect. Current biomarkers for AKI are inadequate for early detection, which is crucial for the initiation of therapeutic strategies at the onset of kidney injury. Presently, kidney transplantation is the only scenario that approximates an experimental AKI ischemia-reperfusion model where such strategies can be tested. Furthermore, conducting rigorous randomized controlled trials is hampered by the difficulty in identifying and selecting high-risk patients who would most benefit from precise interventions. This underscores the urgent need for the discovery of dependable biomarkers and the classification of patients through biological profiling to tailor interventions effectively.

## 6. Conclusions

Chronic Cardiovascular and Kidney Disorder is intricately influenced by the “butterfly effect”, wherein acute injuries in the heart or kidneys can trigger far-reaching consequences in the other organ (Figure 5). This complex interplay is pivotal in understanding CCKD’s pathogenesis and progression. Despite advancements in elucidating these mechanisms, our knowledge remains limited, particularly regarding acute interventions and their long-term effects. Promising therapies like RAAS inhibitors, SGLT2 inhibitors, and GLP1 agonists offer hope in mitigating CCKD progression. Future research should focus on refining these treatments and exploring novel pathways like Galectin-3 inhibition or IL33/ST2 modulation to enhance our therapeutic arsenal against CCKD.

## Figures and Tables

**Figure 1 diagnostics-14-00463-f001:**
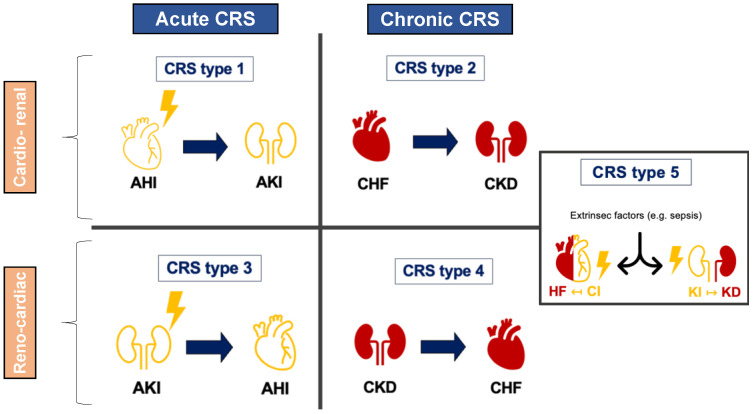
The traditional bidirectional definition and subtypes of cardiorenal syndrome established since 2008. Acute cardiorenal syndrome (CRS) types 1 and 3 arise when an acute heart (AHI) or acute kidney injury (AKI) precipitates an acute injury in the other organ (kidney or heart, respectively). In contrast, chronic CRS types 2 and 4 are characterized by chronic heart failure (CHF) or chronic kidney disease (CKD), which are further complicated by the development of CKD or heart failure (HF), respectively. CRS type 5 refers to scenarios where an external factor causes simultaneous failure in both the heart and kidneys.

**Figure 3 diagnostics-14-00463-f003:**
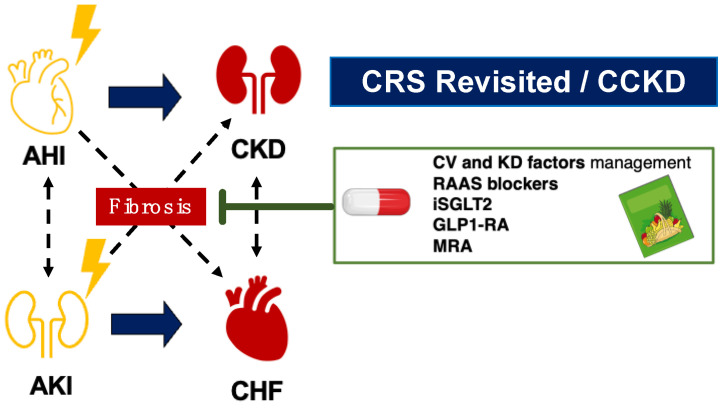
Acute kidney/cardiac injury: the activation of detrimental cardiorenal pathways directly responsible for chronic organ dysfunction in the remote organ (heart or kidney, respectively). AHI, acute heart injury; AKI, acute kidney injury; CHF, chronic heart failure; CKD, chronic kidney disease; CRS, cardiorenal syndrome; CCKD, Chronic Cardiovascular and Kidney Disorder.

**Figure 4 diagnostics-14-00463-f004:**
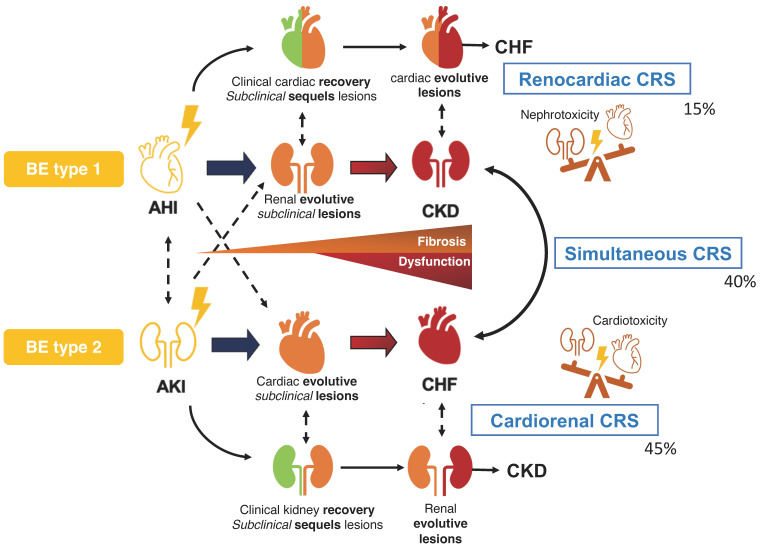
Development of Cardiovascular and Chronic Kidney Disorder through the butterfly effect. In this process, acute heart injury (AHI) or acute kidney injury (AKI), when properly treated, may heal but leave varying degrees of fibrotic damage. These acute incidents activate harmful pathways that lead to significant cardiorenal fibrosis, primarily in the organ opposite the initial injury, resulting in dysfunction (butterfly effect type 1 or 2). Influenced by cardio and nephrotoxic risk factors, the initially impacted organ might cause chronic renal or cardiac dysfunction. Cardiac dysfunction can precede renal dysfunction in cardiorenal syndrome, while renal dysfunction can precede cardiac dysfunction in renocardiac syndrome. In some cases, both dysfunctions may arise simultaneously, termed simultaneous cardiorenal syndrome. BE, butterfly effect; CHF, chronic heart failure; CKD, chronic kidney disease; CRS, cardiorenal syndrome.

**Figure 5 diagnostics-14-00463-f005:**
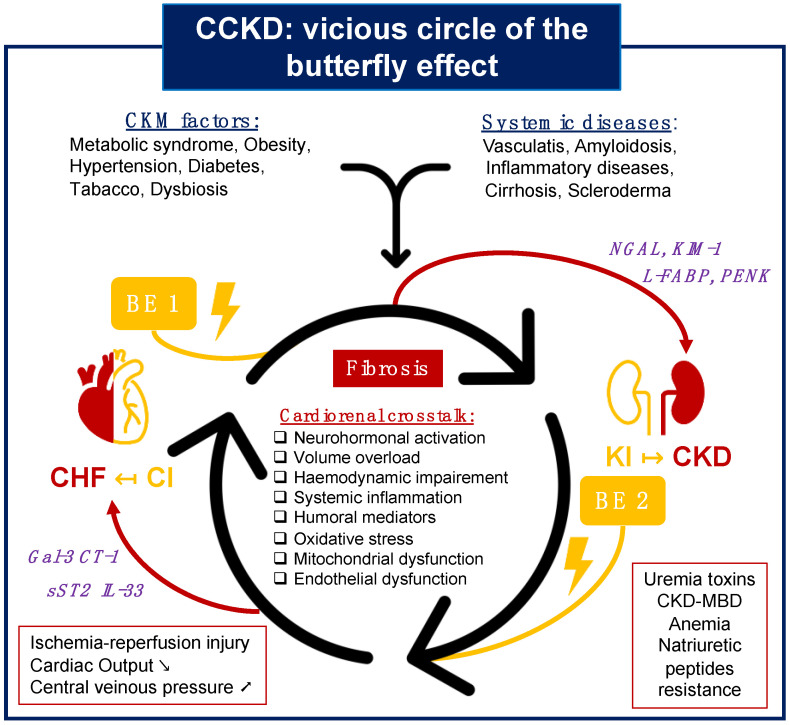
The butterfly effect in CCKD dynamics: pathways to cardiorenal dysfunction. Fibrosis is pivotal in the pathophysiological progression toward cardiac and renal dysfunction, perpetuated by a self-sustaining cardiorenal dialogue involving neurohormonal hyperactivation, hypervolemia, systemic inflammation, oxidative stress, and mitochondrial and endothelial dysfunction. This destructive cycle is further intensified by cardiovascular–metabolic risk factors or specific diseases like systemic inflammatory disorders and amyloidosis. Cardiac dysfunction brings its unique hemodynamic changes, while CKD introduces complications from uremic toxins. The type 1 butterfly effect stems from acute cardiac events, potentially leading to chronic renal failure, with mediators like NGAL, KIM-1, L-FABP, and PENK showing diagnostic and therapeutic potential. Conversely, the type 2 butterfly effect follows acute renal insults, which may precipitate chronic heart failure, with Galectin-3, CT-1, and the IL-33/ST2/sST2 axis as key mediators for diagnosis and treatment. CI: cardiac injury; KI: kidney injury; CCKD: Chronic Cardiovascular and Kidney Disorder; CKM: Cardiovascular–Kidney–Metabolic Disorder Syndrome; CHF: chronic heart failure; CKD: chronic kidney disease; NP: natriuretic peptides; CKD-MBD: CKD–Mineral Bone Disorder; BE: butterfly effect; Gal-3: Galectin-3; CT-1: Cardiotrophin-1; sST2: soluble Suppressor of Tumor-2; IL-33: Interleukin-33; L-FABP: liver fatty acid-binding protein; KIM-1: kidney injury molecule-1; NGAL: Neutrophil Gelatinase-Associated Lipocalin; PENK: Proenkephalin A.

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
