# Peer review of "Unraveling Chronic Cardiovascular and Kidney Disorder through the Butterfly Effect"

_diagnostics, 2024, doi:10.3390/diagnostics14050463_

Round 1
Reviewer 1 Report
Comments and Suggestions for Authors
In this review, Bedo and coworkers review the concept of cardiorenal syndrome in light of the most recent works on the topic and introduce the concept of the butterfly effect. The butterfly effect (BE) theory allows faster action on the affected organs as a continuum assuming the existence of subclinical damage. However, the limitation of the proposed model remains treatment and the effectiveness of treatment in the long term in limiting the progression of organ's damage as a consequence of the primary insult, since the recommended therapy is already applied. Rather, it would be interesting to understand whether with antifibrotic therapy, which is already ongoing , may lead to an icreased protection on a further acute cardiac or renal insult. A part for theabove issue the paper is well written and quite interesting
Comments on the Quality of English LanguageThe manuscript is well written with only minor mistakes, easy to read and interesting.
Reviewer 2 Report
Comments and Suggestions for Authors
"Chronic Cardiovascular Kidney Disorder (CCKD) poses a burgeoning healthcare challenge, characterized by intricate interactions between heart and kidney diseases. This comprehensive review delves into the phenomenon of the 'butterfly effect' within the realm of CCKD, shedding light on the pivotal roles played by acute kidney injury and heart failure. Furthermore, it explores promising therapeutic avenues and identifies novel targets to enhance the efficacy of CCKD management.
The introduced concept of the 'cardiorenal butterfly effect' in this review article is intriguing, offering a fresh perspective on the dynamic interplay between acute insults to the cardiac and renal systems.
Upon thorough evaluation, the review article stands as commendable and is poised for acceptance with only minor recommendations:
- Incorporate a Brief Discussion on Limitations:
- Expand the narrative by including a brief discussion on the limitations and potential challenges associated with applying the 'cardiorenal butterfly effect' concept in clinical practice. This addition will provide depth and realism to the proposed model.
- Ensure Consistency in Font Usage:
- Homogenize the font used in both the main text and comments for a uniform presentation. This adjustment will enhance the overall visual coherence of the document.
- Address Empty Rectangle in Figure 2:
- Investigate and rectify the empty rectangle in the upper left corner of Figure 2. Confirm if any text is missing from this area and make the necessary adjustments.
